# Predicting Office Workers’ Productivity: A Machine Learning Approach Integrating Physiological, Behavioral, and Psychological Indicators

**DOI:** 10.3390/s23218694

**Published:** 2023-10-25

**Authors:** Mohamad Awada, Burcin Becerik-Gerber, Gale Lucas, Shawn C. Roll

**Affiliations:** 1Department of Civil and Environmental Engineering, University of Southern California, Los Angeles, CA 90089, USA; becerik@usc.edu; 2USC Institute for Creative Technologies, University of Southern California, Los Angeles, CA 90089, USA; lucas@ict.usc.edu; 3Chan Division of Occupational Science and Occupational Therapy, University of Southern California, Los Angeles, CA 90089, USA; sroll@usc.edu

**Keywords:** productivity, stress, mood, eustress, distress, psychological state, physiological features, behavioral features

## Abstract

This research pioneers the application of a machine learning framework to predict the perceived productivity of office workers using physiological, behavioral, and psychological features. Two approaches were compared: the baseline model, predicting productivity based on physiological and behavioral characteristics, and the extended model, incorporating predictions of psychological states such as stress, eustress, distress, and mood. Various machine learning models were utilized and compared to assess their predictive accuracy for psychological states and productivity, with XGBoost emerging as the top performer. The extended model outperformed the baseline model, achieving an R^2^ of 0.60 and a lower MAE of 10.52, compared to the baseline model’s R^2^ of 0.48 and MAE of 16.62. The extended model’s feature importance analysis revealed valuable insights into the key predictors of productivity, shedding light on the role of psychological states in the prediction process. Notably, mood and eustress emerged as significant predictors of productivity. Physiological and behavioral features, including skin temperature, electrodermal activity, facial movements, and wrist acceleration, were also identified. Lastly, a comparative analysis revealed that wearable devices (Empatica E4 and H10 Polar) outperformed workstation addons (Kinect camera and computer-usage monitoring application) in predicting productivity, emphasizing the potential utility of wearable devices as an independent tool for assessment of productivity. Implementing the model within smart workstations allows for adaptable environments that boost productivity and overall well-being among office workers.

## 1. Introduction

The contemporary economic landscape has witnessed profound transformations in employment structures, with office work firmly establishing itself as a cornerstone. Office work, one of the most prevalent forms of employment, plays a pivotal role in the economy. In the United States alone, there are approximately 18 million office workers who contribute significantly to the financial returns of companies and the economy as a whole [1]. The productivity of office workers has a direct impact on the success and growth of organizations. High levels of productivity result in increased efficiency, better time management, and a higher quality of work output. This, in turn, leads to increased profits and a positive financial return for companies. On the other hand, a decline in productivity can lead to decreased efficiency, increased costs, and a reduced financial return. Therefore, it is essential for companies to focus on improving the productivity of their office workers to drive their success and maintain higher job satisfaction, improved performance, and increased overall well-being among office workers [2].

As businesses grapple with rapid technological advancements, evolving global challenges, and a shift in work patterns—most notably the rise in remote working in the wake of the COVID-19 pandemic—the traditional metrics and methods for monitoring productivity increasingly seem outdated. The inherent subjectivity and episodic nature of these methods, such as direct observations and performance reviews, introduce potential inconsistencies and biases in their assessments [3,4]. Moreover, as the dynamics of work evolve, so does the need for a more holistic understanding of productivity that encompasses not just the tasks completed but the well-being and psychological state of the employee.

Modern wearable devices and software tools enable continuous capture and analysis of physiological and behavioral data, uncovering insights that might be overlooked by traditional methods. Typically, such physiological and behavioral data have been used to infer an individual’s psychological state, with much of this research centered on predicting stress and mood [5]. However, given the established connection between psychological well-being and productivity [6,7], these features can be incorporated into a comprehensive machine learning (ML) framework to holistically monitor productivity, factoring in the psychological state of employees.

Interestingly, to date, there is an obvious absence of research that specifically explores the use of physiological and behavioral features as predictors of productivity. Additionally, given the well-established relationship between psychological states and productivity, it becomes appropriate to consider whether the incorporation of these states could enhance the accuracy of productivity predictions. The underlying premise here is twofold. First, while physiological and behavioral features might offer a direct understanding of productivity, the addition of psychological states provides a more layered and comprehensive perspective. Second, productivity might not only be directly influenced by physiological and behavioral features but also indirectly shaped by the psychological states of workers.

In essence, this study seeks to address three research questions:What level of prediction accuracy is attainable when focusing solely on the physiological and behavioral features to forecast the productivity of office workers?How does the inclusion and prediction of the psychological state of office workers, when combined with physiological and behavioral features, enhance the precision of productivity prediction?How do different modalities, specifically wearable devices for physiological monitoring versus workstation addons for behavioral data, compare in their effectiveness and accuracy in predicting productivity?

This paper is structured as follows: In Section 2, an overview of related work is presented. Section 3 presents details about the process of collecting data, including the experimental setup, techniques for data cleaning and processing, and the training and evaluation of various ML algorithms. Section 4 provides an overview of the results along with their discussion. Finally, Section 5 and Section 6 conclude this paper by summarizing the findings, pointing out the limitations of our study, and offering suggestions for future research.

## 2. Related Work

### 2.1. Traditional Methods for Productivity Assessment

Traditional methods for assessing employee performance and productivity include direct observation, performance reviews, and task completion rates [3,4,8]. Direct observation involves supervisors monitoring employees to gauge productivity levels, serving as a fundamental technique in various managerial styles. Performance reviews are routine evaluations conducted by superiors, blending quantitative and qualitative data to assess employee contributions against set objectives, or KPIs. Task completion rates focus on quantifying productivity by measuring the number or percentage of tasks completed within a set time frame. These traditional methods are valued for their simplicity, cost-effectiveness, and human element [3,4]. They do not require specialized tools, making them accessible for businesses of all sizes. Additionally, they capture subtle aspects of performance that may be overlooked by purely numerical data, such as teamwork and initiative observed during direct evaluation.

However, there are drawbacks. Traditional methods provide episodic rather than continuous insights, failing to capture sustained performance trends. In addition, with the rise in remote work, the need for autonomous assessment tools has grown as methods like direct observation become harder to implement. Furthermore, while task completion rates provide quantitative data, they do not consider the psychological factors impacting productivity, offering only a numerical output without insight into the emotional and mental states affecting performance.

### 2.2. Physiological and Behavioral Features for Productivity Assessment

Given the limitations of conventional productivity assessment techniques, ML emerges as a promising solution. ML algorithms facilitate ongoing monitoring and assessment of productivity by uncovering patterns across a variety of data sources over prolonged intervals. One of the standout advantages of ML is its adeptness at pinpointing “subtle performance fluctuations”, which refers to the minor yet often significant variations in an employee’s productivity that might otherwise go unnoticed with traditional methods. These fluctuations can be indicative of emerging patterns, potential burnouts, or adaptability challenges that an employee might be facing [9]. ML provides timely insights into nuanced performance shifts, enabling early detection and intervention to ensure sustained productivity and well-being. Another fundamental strength of ML is its capability to assess diverse performance aspects, surpassing the episodic nature of traditional techniques. However, the efficacy of ML in this domain is contingent upon the cautious selection of relevant features.

Among the features attracting attention in this context, physiological metrics, behavioral attributes, and computer interaction patterns stand out [10]. Yet, comprehensive research examining their direct association with productivity is sparse. Through the evolution of wearable technology, devices like smartwatches now offer widespread access to physiological metrics, including but not limited to heart rate (HR), electrodermal activity (EDA), and skin temperature (ST). Although the direct implications of these metrics for productivity assessment remain largely uncharted, numerous studies have associated physiological signals with mental workload, focus, and overall well-being—elements inherently linked to productivity [11,12]. For example, a study by Zahmat Doost and Zhang [13] revealed that interruptions at work can increase mental workload, strain cognitive capacities, reduce productivity, and correlate with higher EDA and elevated HR. This underscores the promising potential of harnessing physiological signals as reliable predictors of productivity.

Behavioral attributes present a broader range of indicators. Software tools can record active work periods, application engagement, and peripheral usage like mouse and keyboard interactions. Additionally, technologies, such as workstation-mounted cameras, have the potential to offer granular insights into elements like facial expressions and posture during tasks. Although research directly linking these attributes to productivity is limited, their potential significance in understanding an individual’s work engagement or frustration indicates a possible correlation between behavioral characteristics and productivity. For example, Whitehill et al. [14] discovered a strong correlation between head rotation and upper-lip raising and work engagement. In contrast, Grafsgaard et al. [15] suggested that lowered brows might indicate frustration, leading to decreased productivity. Meanwhile, Babaei et al. [16] identified gaze movements and gaze angle as signs of challenging work, which implies the need for additional effort, focus, and productivity to address these challenges. These findings underscore the potential of behavioral attributes as nuanced indicators, pointing towards their utility in predicting and understanding productivity.

In summary, the utilization of physiological and behavioral features for productivity assessment is a promising approach to address the limitations of conventional methods. Machine learning excels at uncovering subtle performance fluctuations, potentially indicating emerging patterns, burnout risks, or adaptability challenges. While research directly linking these features to productivity is limited, their associations with factors like mental workload, engagement, and frustration highlight their potential value in predicting and understanding office worker productivity. Despite the absence of a well-established ML framework combining these features, their indirect associations with productivity offer a compelling rationale for further exploration in enhancing workplace productivity assessment.

### 2.3. Psychological States for Productivity Assessment

A psychological state refers to an individual’s emotional, cognitive, and motivational condition at any given moment [17]. The relationship between psychological states and productivity has been an area of keen interest in recent years. Numerous studies have sought to explore the intricate linkages between different psychological states and their potential impact on productivity. Central to this exploration is a nuanced understanding of stress and its two primary manifestations: eustress and distress. Stress, when experienced in moderation, often falls under the bracket of eustress, the “beneficial” or positive form of stress. Eustress, characterized by a balance of stimulation and challenge, is known to boost productivity. It does so by enhancing cognitive performance, fostering resilient coping strategies, and steering behavior in a goal-oriented direction [18]. However, when the threshold of what is considered “moderate” stress is crossed, individuals shift from experiencing eustress to distress. Distress, a form of stress perceived as negative, hampers productivity by impairing cognitive functions and obstructing efficient task completion. It exemplifies the adverse outcomes that stem from excessive or overwhelming stress [18]. The same external factor could lead to either eustress or distress, depending on individual perception, resilience, and coping mechanisms. Beyond stress, the role of mood in influencing productivity cannot be understated. Research underscores a robust correlation where positive moods amplify productivity levels, while negative moods tend to dampen them [19].

Traditionally, psychological states, including stress, eustress, distress, and mood, have been measured through self-report scales that allow for subjective evaluation by the individuals themselves. For instance, stress levels are commonly assessed using standardized instruments like the State Anxiety Inventory [20] or through direct self-report measures where individuals are asked to evaluate their own levels of stress [21]. When assessing eustress and distress, researchers frequently use the Valencia Eustress–Distress Appraisal Scale (VEDAS). The VEDAS has undergone validation in numerous research studies and has demonstrated efficacy in assessing distress and eustress [22,23]. This instrument utilizes a six-point Likert scale, where participants evaluate their perception of work as a source of challenge (indicative of eustress) or pressure (indicative of distress). In terms of mood assessment, the Positive and Negative Affect Schedule (PANAS) [24] is a well-established tool, although there are also studies in which mood is assessed through straightforward, direct questions [25].

These self-report methods are typically used as ground truth in the establishment of ML-related models. There have been a plethora of research papers that have implemented ML models using physiological and behavioral data to predict the stress and mood levels of office workers [26,27]. For example, Koldjik et al. [21] analyzed data like EDA, heart rate variability (HRV), facial expressions, computer interactions, and posture from 25 participants. Their models impressively predicted stress levels with 90% accuracy and similarly discerned emotional valence and arousal. In the realm of mood predictions, Shu et al. [28] achieved an 84% accuracy rate in classifying happy, sad, and neutral moods using HR data from wristbands. Meanwhile, Narayana et al. [29] attained an 89% accuracy in predicting moods using facial features alone. Additionally, Li et al. [30] harnessed computer usage along with HR and HRV data from 7 participants, achieving a 71% prediction accuracy for eustress. Despite significant advances and high accuracy rates, a gap remains in the literature. Current models effectively predict stress and mood levels but are not part of broader assessments of employee productivity.

Physiological and behavioral features currently play a role in evaluating office workers’ psychological states. Given their potential for predicting productivity and recognizing the established link between psychological conditions and productivity, there is a strong case for introducing a comprehensive ML framework. This framework could integrate predictions of mood, stress, eustress, and distress alongside physiological and behavioral attributes to enhance productivity prediction accuracy.

## 3. Methodology

We performed an experimental study to predict individuals’ perceptions of stress, eustress, distress, and mood, which are utilized to gauge their perceived productivity levels. To capture a broad spectrum of signals from each participant, our 70 min experiment consisted of two phases: the first phase involved low-stress engagement with a computer workstation, followed by a second phase that incorporated several stressful elements. The aim of this experimental method was to create diverse work conditions that could impact the emotional states of participants and, subsequently, their productivity.

### 3.1. Participants

This research included 48 volunteers who took part willingly (28 women and 20 men). The participants were a mix of graduate and undergraduate students, averaging 22.6 years old with a standard deviation of 2.1 years. Individuals with vision problems that could hinder computer-related tasks, psychological disorders that make them more vulnerable to stress, pregnant women, and those using medication that might impact physiological signals were excluded from this study. Informed consent was obtained from all subjects involved in this study.

### 3.2. Data Collection

During our study, the participants were equipped with two types of sensors: an E4 Empatica wristband [31] and an H10 Polar chest strap [32]. These sensors were employed to gather various physiological information, including HR, EDA, ST, blood volume pulse (BVP), and wrist accelerations in the x, y, and z directions. Additionally, a Microsoft Azure Kinect DK camera [33] was positioned to capture participants’ facial expressions throughout the experiment. To record their computer interactions, such as keystrokes and mouse clicks, a logging application called Mini Mouse Macro [34] was utilized. Figure 1 illustrates the configuration of the workstation used in the experiment.

The experimental procedure comprised two distinct phases: low-stress work and high-stress work. At the commencement of each phase, participants were instructed to remain motionless for a duration of 5 min to record their baseline physiological data while at rest. Following these initial resting periods, participants were required to rate their baseline subjective stress and mood levels on a scale ranging from 0 to 100. A rating of 0 denoted the absence of stress or a negative mood, whereas a rating of 100 indicated the presence of extreme stress or a positive mood. During both phases, at intervals of 5 min, participants were presented with a pop-up questionnaire on the computer screen. This questionnaire prompted them to rate their perceived stress and mood levels using the 0–100 scale. Furthermore, participants were requested to evaluate the nature of their work experience as either eustress or distress using the VEDAS. Eustress was evaluated based on the perception of opportunity or challenge, employing a 6-point scale ranging from “very definitely is not” to “very definitely is”. Similarly, distress was assessed as a source of pressure using the same 6-point scale. Additionally, participants were asked to rate their perceived productivity on a scale of 0 to 100, with 0 representing an extreme lack of productivity and 100 indicating an exceptional level of productivity. The sequence of the two phases was kept consistent for all participants, beginning with the low-stress tasks followed by the high-stress tasks. Randomization was not employed in this study, as its primary focus was on data collection rather than examining the effects of varying task sequences. Prior to the commencement of the experiment, participants were provided with guidelines and definitions to help them understand and interpret the scales accurately. For instance, for stress, a score of 0 was explained as feeling completely at ease and relaxed, whereas a score of 100 represented feeling overwhelmed and unable to cope with the pressure. Analogously, mood ratings were clarified, with 0 representing feelings of sadness or frustration and 100 reflecting feelings of happiness or excitement.

The objective of the low-stress phase was to provide participants with a perception of autonomy and minimize external demands or pressures associated with the task. In contrast, the high-stress phase was deliberately designed to introduce a workload that was challenging within a restricted timeframe. During the low-stress phase, participants were allotted 40 min to construct a slide deck centered around a topic of their preference, such as a beloved book, television series, or movie. Subsequently, participants were granted a break before resuming the experiment. This break was intentionally brief, limited to no more than two minutes, in order to preserve the participants’ focus and work-oriented mindset.

Upon returning and recording their resting physiological data, participants were informed that they would no longer be working on their initial topic. Instead, they were instructed to engage with an unfamiliar topic for a duration of 30 min. To identify a topic that would provoke a high-stress response, we conducted a pilot study, testing various themes. Ultimately, we settled on a discussion regarding the scientific and philosophical contributions of two ancient Greek philosophers and the enduring impact of their ideas on contemporary human life.

In the experiment’s high-stress condition, additional stressors were applied to elevate pressure. Participants had to activate video cameras and share screens via Zoom, with a confederate posing as a professor specializing in optimal work environments for productivity. This authoritative figure was introduced to increase expectations and induce performance anxiety and stress due to fear of negative evaluations from someone significant in the field. The confederate stated they would monitor and potentially decrease participants’ scores based on performance, simulating a high-pressure work environment with critical evaluations. A computer application displayed fluctuating ratings for the professor throughout the 40 min task, with these manipulations standardized across participants. Participants were informed their scores would be compared to others’, introducing a competitive element and inducing stress through fear of inadequacy or failure. Compensation varied based on performance, with the highest-scoring participants receiving USD 50 and the lowest USD 5. This variable compensation introduced financial stressors, playing on loss aversion and stress associated with potential monetary loss. However, during debriefing, participants learned the confederate was not an actual expert, and their scores did not affect compensation. All participants received maximum compensation regardless of performance.

### 3.3. Data Processing

The participants’ perceived stress and mood ratings were calculated by subtracting their ratings during the resting period from the corresponding ratings during the experimental phases. This resulted in perceived stress and mood ratings ranging from 0 to 100, representing the deviation from the baseline. The appraisals of eustress and distress were transformed into a binary outcome. Responses falling within the categories of “very definitely is not a source of”, “definitely is not a source of”, and “generally is not a source of” were combined to create a category indicating that the stress was not appraised as either eustress or distress. Similarly, the categories of “very definitely is a source of”, “definitely is a source of”, and “generally is a source of” were merged to form a category indicating that the stress was appraised as either eustress or distress. Given that participants were not engaged in any work tasks during the baseline phases, the perceived productivity baseline was assumed to be 0, reflecting the absence of any productivity during those periods.

The data collected in this study were segmented into 30 s time windows to extract physiological and behavioral features. This choice was informed by Bernardes et al.’s research [35], which demonstrated that 30 s windows provided the smallest reliable timeframe for extracting HRV features that accurately assess psychological states. To analyze HRV and obtain time- and frequency-domain indices of the HR signal at each 30 s interval, the Kubios software [36] was employed. A moderate artifact correction technique was applied to pinpoint R-R intervals that deviated more than 0.25 s from the mean. This approach retained the variability of the data while managing any existing artifacts. Furthermore, Kubios incorporated a piecewise cubic-spline interpolation process to fill in flawed or missing data, ensuring a more refined and precise HRV reading.

Data gathered from the Empatica E4 underwent processing prior to the extraction of features to minimize noise, akin to the methodology adopted in an earlier study [37]. BVP and ST signals were refined using winsorization [38], a statistical method that removes outliers beyond the 2nd and 98th percentiles. For processing the EDA signal, we employed the MATLAB Ledalab toolbox [39]. This included the use of a Butterworth low-pass filter, Hanning smoothing encompassing a span of 4 consecutive data points, and manual artifact rectification to eliminate any noise potentially due to motion or other external disruptions. After this cleaning phase, we computed the average, variance, median, minimum, maximum, 25th and 75th percentiles, and the fitted slope (i.e., linear regression slope) of BVP, EDA, and ST to provide a comprehensive evaluation of the multiple facets inherent in psychological assessment [26].

The OpenFace tool [40] was utilized to retrieve the mean and standard deviation values of facial action unit intensities, gaze angles, and head movement and orientation at 30 s intervals, derived from the RGB video recorded by the Kinect camera. Furthermore, keyboard strokes and mouse clicks were recorded for each 30 s time window. Finally, the mean and standard deviation of the wrist acceleration in the x, y, and z planes were recorded. A summary of the feature dataset, including the various measures and statistics derived from the different physiological and behavioral sources, can be found in Table 1.

The final dataset utilized in this study consisted of 6720 data instances, collected from 48 participants. Each participant contributed 70 min of data, with 2 instances per minute using 30 s time windows. The final dataset comprised 83 features, encompassing 34 physiological features, 48 behavioral features (including 3 related to human–computer interactions, 39 facial-related features, and 6 hand-wrist acceleration features), and a feature indicating the participant’s gender. To facilitate proper normalization, robust scaling was applied on a per-participant basis to ensure consistency across the dataset.

### 3.4. Analysis Plan

Our analysis consisted of four distinct steps. In the first step, we focused on predicting productivity using the physiological and behavioral dataset solely. In the second step, we extended our productivity prediction model by incorporating the predictions of psychological states (stress, mood, eustress, and distress) into the dataset. To achieve this, we employed various ML algorithms and selected the best-performing one for each outcome under study. Continuous metrics such as stress, mood, and productivity are predicted using regression models, including linear regression, ridge regression, lasso regression, random forest, gradient boosting regressor, and the Extreme Gradient Boosting (XGBoost) regressor. On the other hand, binary outcomes like eustress and distress require classification models, and we evaluate algorithms such as logistic regression, random forest, gradient boosting classifier, XGBoost classifier, decision tree, and Support Vector Classifier (SVC). To evaluate the performance of these ML models, we employed an 80–20% split, where 80% of the dataset was utilized for training and the remaining 20% for testing purposes. To assess the regression models, we considered the Mean Absolute Error (MAE) and R-squared (R^2^) as evaluation metrics. For the classification models, we utilized accuracy and the F_1_-score to evaluate their performance. In the third step, a feature importance analysis was conducted on the extended productivity model. This process aimed to elucidate significant predictors of productivity by calculating importance scores for each feature, utilizing the model’s feature importance attribute. This analysis identified and examined the top 15 features influencing productivity predictions, aiding in understanding the model’s predictive mechanisms. In the fourth step, we evaluated the effectiveness of physiological and behavioral features individually in predicting productivity. Figure 2 presents a summary of the analysis.

## 4. Results and Discussion

### 4.1. Predicting Mood, Stress, Eustress, and Distress

In this study, we conducted a regression analysis to evaluate the performance of various algorithms in predicting stress and mood levels. The regression analysis aimed to assess the effectiveness of each model in capturing the underlying patterns and relationships. Additionally, a classification analysis was performed to determine the accuracy of different models in predicting eustress and distress. The classification analysis focused on evaluating the models’ ability to classify individuals into the appropriate stress categories. Table 2 provides a summary of these results.

Linear regression models are ineffective in predicting mood and stress levels, with weak correlations (R^2^: 0.06–0.09) and high average deviations (MAE: 10.36–15.68). In contrast, tree-based ensemble models (random forest, gradient boosting, and XGBoost) outperform linear regression, showing stronger correlations (R^2^: 0.31–0.44) and lower average deviations (MAE: 7.91–12.74). These results indicate that tree-based models better capture the complexities and nuances in mood and stress data.

Among the tree-based models, XGBoost and random forest excel at predicting mood and stress, respectively. Their superior performance can be attributed to the employment of ensemble techniques that leverage multiple decision trees to capture intricate relationships and interactions within the data. These models are adept at handling nonlinearities, outliers, and high-dimensional feature spaces, allowing them to effectively capture the nuanced aspects present in mood and stress data.

When considering eustress and distress predictions, logistic regression shows moderate accuracy (0.79) and F_1_-scores (0.65) for eustress, indicating reasonably accurate predictions. However, its performance is less satisfactory for distress, with lower accuracy (0.53) and F_1_-score (0.68). This limitation can be attributed to the linear nature of logistic regression, which hinders its ability to capture the complexity and nonlinearity associated with distress instances.

The decision tree model performs satisfactorily for eustress (accuracy: 0.74, F_1_-score: 0.81), indicating accurate predictions and a good balance between precision and recall. However, its performance for distress (accuracy: 0.67, F_1_-score: 0.69) suggests a relatively lower ability to accurately classify distress instances, possibly due to overfitting and capturing noise or idiosyncratic patterns that do not generalize well. The Support Vector Classifier exhibits inferior performance for both eustress (accuracy: 0.65, F_1_-score: 0.78) and distress (accuracy: 0.68, F_1_-score: 0.51), indicating less accurate predictions and imbalanced precision and recall. This can be attributed to the sensitivity of the Support Vector Classifier to feature scaling and hyperparameter selection, making it less effective when there is overlap between eustress and distress instances.

Gradient boosting demonstrates strong performance for both eustress (accuracy: 0.79, F_1_-score: 0.86) and distress (accuracy: 0.75, F_1_-score: 0.77), capturing complex relationships effectively. Random forest performs even better with higher accuracy for both eustress (accuracy: 0.84, F_1_-score: 0.88) and distress (accuracy: 0.81, F_1_-score: 0.82), integrating multiple decision trees to capture a wider range of patterns. XGBoost emerges as the top performer, achieving the highest accuracy for both eustress (accuracy: 0.88, F_1_-score: 0.91) and distress (accuracy: 0.85, F_1_-score: 0.85), utilizing advanced ensemble techniques, regularization, and optimization strategies to handle complex datasets effectively.

Comparing our models with those from the literature is challenging due to variances in data signals, features, ML algorithms, and experimental conditions. Nevertheless, our top-performing models appear competitive. For instance, a study by Yu et al. [41] utilized wrist acceleration, EDA, and ST from mobile phone data to predict mood and stress on a 0–100 scale, achieving MAEs of 13.7 and 12.8 for stress and mood prediction, respectively. In comparison, our models achieved MAE values of 7.91 and 9.81. In a separate study, Li et al. [30] combined computer usage with HR and HRV data, achieving a prediction accuracy of 71% for eustress, whereas our model reported an accuracy of 88%. Additionally, a study focused on detecting distress events using an ML model based on EDA and BVP attained an F_1_-score of 0.71 [42], compared to our model’s F_1_-score of 0.85.

### 4.2. Baseline versus Extended Productivity Models

We begin by establishing the baseline productivity model, which predicts productivity solely using the physiological and behavioral features collected during the experiment. Subsequently, the extended productivity model incorporates additional predictions from the best-performing models of mood, stress, eustress, and distress in addition to those features. Specifically, we utilized the XGBoost models outlined in Table 2 for predicting mood, eustress, and distress. For stress prediction, we employed the random forest model, also presented in Table 2. We tested several ML algorithms, and the outcomes are presented in Table 3. It is noteworthy to highlight that in the extended model, stress, mood, eustress, and distress were already predicted using the physiological and behavioral features. Consequently, we conducted a thorough assessment to address the potential issue of multicollinearity, employing the well-established statistical measure known as Variance Inflation Factor (VIF). Multicollinearity arises when two or more independent variables within a regression model exhibit high correlation, leading to undesirable consequences such as unstable and unreliable coefficient estimates. VIF precisely quantifies the extent to which the variance of an estimated regression coefficient is inflated due to multicollinearity. Through our analysis, we ascertained that the correlation levels were moderate, with VIF values ranging from 1 to 8. This finding indicates that the multicollinearity in the extended model is not severe and does not necessitate any alterations.

The productivity regression analysis highlighted the consistent superiority of the extended model over the baseline model in predicting productivity. Remarkably, XGBoost was the top-performing algorithm for both models. In the comparison, the extended model posted an impressive R^2^ of 0.60 and a lower MAE of 10.52, against the baseline’s R^2^ of 0.48 and MAE of 16.62. Such improved performance was consistent across various algorithms, substantiating the extended model’s enhanced effectiveness. While the extended model demonstrates a noteworthy improvement over the baseline, it is important to acknowledge that an R^2^ value of 0.60 represents a moderate correlation. This indicates that, despite its superior performance, the extended model still has room for refinement to capture the complexities of productivity more accurately.

What set the extended model apart was its integration of additional features. Instead of relying solely on physiological and behavioral variables like the baseline model, it incorporated predictions from leading models on mood, stress, eustress, and distress. This broadened feature set ensured a more in-depth understanding, capturing the intricate human conditions that influence productivity. The stark difference in performance metrics, particularly the R^2^ improvement and MAE reduction for XGBoost, attests to the potency of a holistic approach. It drives home the point that for nuanced, human-centric predictions like productivity, it is imperative to embrace a wider spectrum of influencing factors.

It is noteworthy that the direct use of self-reported scores (as opposed to predictions) for stress, mood, eustress, and distress in the extended productivity model yielded a performance enhancement. Specifically, improvements ranged from 0.7% to 2.4% when compared to the results obtained from the extended predictive model based on stress, mood, eustress, and distress predictions, as presented in Table 3. Utilizing a model based on psychological state predictions may necessitate initial ground-truth data for training the ML model, after which user input becomes unnecessary. On the other hand, relying solely on self-reported metrics would require continuous user input. This could compromise the objective of establishing an automated framework, as it may result in frequent interruptions to work activities. Consequently, it may be judicious to forgo the slight increase in prediction accuracy in favor of the operational advantages offered by a fully automated ML system built on psychological state predictions.

As far as our knowledge extends, there has been no prior investigation into the utilization of physiological or behavioral features to evaluate productivity among office workers, let alone the incorporation of psychological states. Thus, these findings provide an essential basis for future endeavors focused on the development of machine learning-driven solutions to predict and monitor productivity in an office setting.

### 4.3. Analyzing Feature Importance

To gain a deeper understanding of the prediction mechanism employed by the extended model, we conducted a feature importance analysis. This analysis aimed to reveal the physiological and behavioral features that exerted the greatest influence in predicting productivity, as well as shed light on the role of emotional states in this prediction process. We used the feature importance attribute of the model to measure how much each feature contributed to the overall prediction accuracy. An importance score for each feature was calculated to assess the contribution of each feature to the overall prediction accuracy of the ML model. The 15 most important features of the extended productivity model are presented in Figure 3.

The analysis of feature importance in predicting productivity revealed several key predictors, including physiological, behavioral, and emotional features. Of particular interest are the emotional states, which emerged as significant contributors to productivity. Understanding why these emotional states play a crucial role in predicting productivity requires a closer examination of their underlying mechanisms. Emotions, as complex and dynamic psychological states, have long been recognized for their influence on cognitive processes and behavior. In the context of productivity, emotions can shape an individual’s motivation, attention, decision-making, and overall cognitive functioning. The appearance of emotional states as important predictors in the extended model highlights their importance in capturing the multifaceted nature of productivity.

One prominent emotional characteristic to consider is predicted eustress, a form of stress that conveys positive implications. Eustress denotes a moderate degree of stress that individuals perceive as advantageous or motivating. It emerges in circumstances where individuals encounter a sense of challenge, excitement, or anticipation. Eustress can heighten cognitive performance, foster adaptive coping mechanisms, and facilitate goal-oriented conduct [43]. By integrating predictions of eustress, the expanded model acknowledges the potential advantages of stress in optimizing productivity. It posits that an optimal level of stimulation and challenge can cultivate engagement, concentration, and the mobilization of cognitive resources. Findings from the *t*-test provide support for this assertion (t(6718) = 3.95, *p* = 0.04); the projected productivity level was higher (42.75 ± 27.05) when the model predicted eustress compared to scenarios without eustress (29.17 ± 26.32).

Among the emotional features under investigation, predicted mood emerged as a notably influential factor in the prediction of productivity. Mood encompasses an individual’s overall emotional state, encompassing various degrees of positivity, negativity, or neutrality. Positive mood states, characterized by emotions such as enthusiasm, joy, and contentment, have been consistently linked to heightened cognitive flexibility, enhanced creative thinking, and improved problem-solving capabilities [19]. Such positive affective experiences contribute to increased motivation, active engagement in tasks, and effective information processing. On the other hand, negative mood states, including emotions such as sadness, anxiety, or frustration, can detrimentally affect cognitive functioning, leading to decreased levels of productivity. A correlation analysis revealed a statistically significant yet weak positive Pearson correlation between mood and productivity (r = 0.10, N = 6720, *p* < 0.001). The extended model’s ability to capture and integrate these mood predictions enables a more comprehensive understanding of the impact of affective experiences on productivity.

In our study, we initially hypothesized that distress would have a negative impact on productivity. However, contrary to our expectations, our findings revealed a different pattern. The results of our *t*-test analysis contradicted the expected trend, as individuals experiencing distress (41.56 ± 25.72) reported higher levels of productivity compared to those without distress (34.73 ± 29.01). To gain a deeper understanding of this unexpected outcome, we further investigated how the simultaneous presence of eustress and distress influenced individuals’ perceptions of productivity. Specifically, we examined four combinations of predictive outcomes: “no-eustress” and “no-distress”; “eustress” but “no-distress”; “eustress” and “distress”; and “no-eustress” but “distress”.

Significant statistical differences were observed in the predicted productivity levels (F(3, 6716) = 142.30, *p* ≤ 0.001). These findings suggest that the interplay between eustress and distress, rather than the presence of distress alone, may have a nuanced impact on individuals’ productivity levels. Specifically, when both “no-eustress” and “no-distress” were simultaneously predicted by the model, the average predicted productivity level was found to be M = 27.52 ± 26.37. In cases where the model predicted “eustress” but “no-distress”, the average predicted productivity level was M = 45.95 ± 29.73. When both “eustress” and “distress” were predicted by the model, the average predicted productivity level was M = 41.92 ± 25.84. Conversely, when the model predicted “no-eustress” but “distress”, the average predicted productivity level was M = 38.21 ± 24.78. These results indicate that the presence of pure distress can lead to decreased predicted productivity compared to a state of pure eustress. However, the lowest predicted productivity level was observed when the model predicted a worker’s state of no eustress and no distress, which may signify a state of boredom or disengagement from work [44].

This study’s results should be interpreted cautiously, as the sample exclusively consists of young students, whose stress responses and perceptions of productivity may not generalize to wider or diverse populations. Their unique academic stressors, coping mechanisms, and perhaps elevated resilience to distress might lead to different productivity outcomes compared to non-student groups.

It is important to acknowledge that emotional states exhibit considerable predictive power in the expanded model. However, it is crucial to consider them in conjunction with physiological and behavioral characteristics. This is because emotional states, while influential, do not exist in isolation; they are intertwined with our physiological responses and the actions we take. By adopting this holistic approach, we gain a more comprehensive understanding of how these factors collectively impact productivity. Table 4 presents the correlation analysis between predicted productivity and the most significant physiological and behavioral features in the predictive framework. In our analysis, we conducted Shapiro–Wilk tests, and all obtained *p*-values exceeded the significance threshold of 0.05, confirming that the assumption of normality for our data is met and validating the use of correlation analysis.

The observation of a positive relationship between ST and productivity suggests a potential interconnection between physiological and cognitive processes. One plausible explanation for this phenomenon lies in the amplified blood flow and metabolic activity that occur during engaging and productive tasks [45]. This heightened physiological response facilitates effective heat transfer from the body’s core to the peripheral regions, consequently leading to an elevation in ST. However, it is noteworthy that while four EDA-related features emerged as primary predictors of productivity, only the median of the EDA signal exhibited a statistically significant positive correlation with the predicted productivity. Although the correlation coefficients between the EDA-related features and the predicted productivity are relatively small, ranging from 0.02 to 0.04, they do show a positive association. This finding could be attributed to the activation of the sympathetic nervous system, which occurs during focused attention and heightened arousal in productive tasks. This activation leads to increased sweat secretion, resulting in higher EDA values [45]. These findings lay the foundation for future investigations aimed at elucidating the underlying relationship between EDA and productivity.

Higher wrist acceleration is found to be negatively correlated with productivity. In tasks involving a keyboard and mouse, optimal productivity is often associated with stable, precise, and limited hand movements. Conversely, higher hand acceleration, suggesting excessive or erratic hand movement, tends to be negatively correlated with productivity as it might indicate non-focused or inefficient activity [46]. Conversely, a positive correlation exists between the standard deviation of AU06 Cheek Raiser and AU10 Upper-Lip Raiser and productivity, suggesting that greater variation in the movements of these action units is associated with higher productivity. This may be attributed to increased facial expressiveness, reflecting active engagement and emotional responsiveness, as well as lower stress levels [47], thus contributing to enhanced productivity. Lastly, a negative correlation is found between the mean of AU04 Brow Lowerer and productivity, suggesting that higher levels of brow-lowering movements are associated with lower productivity. Brow lowering often accompanies negative emotions or concentration, potentially indicating increased cognitive load or negative affect, which may impede productivity [47].

The mean of the head rotation (*z*-axis) feature showed a positive correlation coefficient of 0.06. This finding suggests that greater head rotation is associated with higher productivity levels. The increased head rotation may reflect heightened attentiveness and active involvement in tasks, indicating an individual’s active scanning of the environment or engagement in complex cognitive processes. These cognitive processes likely contribute to enhanced productivity by facilitating information processing and task engagement. Additionally, the mean of the gaze angle (*x*-axis) feature demonstrated a notable positive correlation coefficient of 0.13 with predicted productivity. This correlation suggests that a more direct and focused gaze is associated with higher levels of productivity. A concentrated gaze directed toward a task or relevant stimuli signifies sustained attention and cognitive engagement. This focused visual attention is indicative of an individual’s ability to maintain cognitive resources on the task at hand, resulting in improved productivity.

It is crucial to acknowledge that the interpretations put forth are grounded in observed correlations, signifying a relationship between the variables under investigation. However, to advance our understanding and draw more definitive conclusions, further research is warranted to establish causality and unveil the specific mechanisms that underlie the intricate relationships between these features and their impact on productivity. Moreover, it is imperative to recognize the interconnection between these findings and the experimental results conducted in our study. To ensure the generalizability and applicability of these results in real-world settings, a comprehensive longitudinal data collection approach becomes indispensable. By systematically gathering physiological and behavioral data over an extended period of time and employing the ecological momentary assessment method to continuously inquire about participants’ psychological states and productivity, we can enhance the generalizability and robustness of our findings. This approach allows us to capture the dynamic nature of these variables in a real-world context and provides a more holistic view of their influence on productivity. Additionally, such an approach enables us to gain insights into the temporal aspects and potential causal pathways, shedding light on the underlying mechanisms that govern these associations.

### 4.4. Comparison between Different Modalities

In this section, our investigation aimed to explore the impact of various modalities on the prediction of productivity. Specifically, we conducted a comparative analysis between wearable devices, namely the Empatica E4 and the H10 Polar, and workstation addons, specifically the Kinect camera and the Mini Mouse Macro. Our focus was directed towards evaluating the performance of the extended productivity model, as we had previously demonstrated its superior predictive capabilities.

The results from Table 5 reveal that the data collected from the wearable devices exhibited superior predictive capabilities, as indicated by an R^2^ value of 0.56 and an MAE of 12.97. In contrast, the data derived from workstation addons yielded comparatively lower predictive accuracy, with an R^2^ value of 0.50 and an MAE of 15.55. The favorable predictive performance of the model utilizing wearable device data is particularly noteworthy. These results are highlighted further when considering the performance of the model that incorporates all available data streams, which yielded an R^2^ value of 0.60 and an MAE of 10.52 (Table 3). While the combined model achieved slightly better accuracy, the wearable device model’s performance remains comparable, emphasizing its potential as a standalone predictive tool.

The findings unequivocally establish that integrating data from wearable devices into productivity models yields markedly superior predictive outcomes compared to relying solely on workstation addons. The essence of this disparity lies in the precision and granularity of the captured data. The wearable devices, carefully selected for this study, exhibited a remarkable concentration of physiological data, boasting high-frequency sampling rates. For instance, the Empatica E4 [48] meticulously recorded BVP (64 Hz), ST (4 Hz), EDA (4 Hz), and wrist acceleration (32 Hz), whereas the H10 Polar collected heart rate (1 kHz) [49] at near real-time intervals. This high-resolution data allowed us to discern nuanced shifts in an individual’s physiological responses, thus enabling a more accurate productivity assessment.

Conversely, the Kinect camera, operating at a somewhat modest 10 frames per second (fps), while proficient in capturing facial expressions and body movements, may have occasionally missed subtler cues. In particular, minor fluctuations in facial expressions that could indicate nuanced emotional states might not have been entirely captured, potentially limiting the depth of contextual information obtained. Furthermore, human–computer interaction features, such as mouse clicks and keyboard keystrokes, while valuable in understanding participant engagement with computer-based tasks, might not provide the most comprehensive representation of productive work. Participants might have been engaged in cognitively demanding activities, such as reading and processing information on the screen or formulating ideas for written responses, which might not manifest through these interaction metrics but still constitute productive work. Therefore, solely relying on these metrics could underestimate the actual productivity levels of participants during computer-based tasks.

Furthermore, wearable devices allow for unobtrusive data collection without altering participants’ natural work routines, mitigating concerns about participant awareness and potential bias [50]. Lastly, the discreet nature of wearable devices somewhat addresses privacy concerns associated with camera-based systems or tracking applications, ensuring participant comfort and the representation of natural work behaviors.

In practical terms, organizations striving for maximum predictive accuracy in productivity monitoring may choose to invest in a fully equipped workstation setup (wearable devices and workstation addons), as it allows for maximum productivity prediction accuracy. However, recognizing that not all organizations possess the necessary resources—be it financial resources, time, or data infrastructure—to support extensive data collection from workstation addons like Kinect cameras, our research underscores the practicality of utilizing simple wearable devices. These devices offer a cost-effective and efficient alternative for monitoring productivity, allowing organizations to achieve predictive accuracy that compares favorably to the maximum prediction accuracy attainable. This flexibility empowers organizations to tailor their productivity monitoring to suit their unique circumstances, objectives, and resource constraints.

## 5. Limitations and Future Work

While this study provides valuable insights into the predictors and mechanisms of productivity, it is important to acknowledge several limitations. Our research was conducted in a controlled laboratory setting, limiting the generalizability of the findings to real-world office environments. To address this, future research should validate the findings in diverse work contexts. Additionally, the correlational nature of our study restricts the establishment of causality between predictors and productivity outcomes. Longitudinal and experimental designs are needed to uncover causal relationships and underlying mechanisms.

Furthermore, it is imperative to consider the demographic specificity of the studied population as a limitation. The participants primarily comprised young undergraduate and graduate students within a confined age range, possessing distinctive educational backgrounds, high motivation, and elevated cognitive reserves. Future research endeavors should consciously aim to engage a more heterogeneous participant pool, encompassing varied age groups, educational levels, and cognitive reserves, to enhance the generalizability and applicability of the findings in real-world, diverse work settings.

Another limitation is the use of an 80–20% model evaluation approach, which may affect the generalizability of the results. The results obtained using the leave-one-participant-out method and cross-validation did not demonstrate high prediction accuracy compared to the 80–20% split method. This discrepancy in performance could be attributed to the substantial personal variability present within our dataset, thus compromising the generalizability of the findings. For that, future research should explore ways to incorporate individual characteristics, such as age and personality traits, into the prediction model.

Moreover, it is crucial to acknowledge the significance of privacy concerns. Nevertheless, it is worth mentioning that engagement in productivity monitoring programs can be optional, granting employees the freedom to decide whether to participate. Upholding privacy rights and guaranteeing clear and open communication regarding data utilization will be vital when integrating these monitoring systems. The practical implementation and integration of the model within workplace systems should be explored, along with the evaluation of its effectiveness in improving productivity and employee well-being. By addressing these limitations, researchers can advance our understanding of productivity and its management in office environments.

## 6. Conclusions

To the best of our knowledge, this research represents the pioneering application of an ML framework to predict perceived productivity based on physiological and behavioral features in the context of smart workstations. The results showed that incorporating predictions of office workers’ psychological states such as stress arousal, eustress, distress, and mood alongside physiological and behavioral features resulted in improved productivity prediction. The feature importance analysis conducted in this study aimed to uncover the key predictors of productivity and shed light on the role of emotional states in the prediction process. Emotional states emerged as significant contributors, with mood, eustress, and distress playing influential roles. This study also identified important physiological and behavioral features related to productivity, such as ST, EDA, wrist acceleration, facial movements, head rotation, and gaze angle. Finally, a comparative analysis between wearable devices (Empatica E4 and H10 Polar) and workstation addons (Kinect camera and Mini Mouse Macro) showed that data collected from wearable devices outperformed workstation addons in predicting productivity, highlighting the potential value of wearable devices as a standalone tool for productivity assessment.

This research has significant implications for office design and management, specifically enhancing productivity. The extended productivity prediction model, considering emotional states, physiological responses, and behavioral characteristics, enables an office that is aware of workers’ emotional and cognitive states. This allows for an adjustable workspace that dynamically adapts to foster productivity. For instance, integrating a smart workstation with productivity prediction features into office systems enables real-time monitoring and response to workers’ emotional and physiological states. Through intelligent lighting, temperature control, and ambient music, the office environment can be optimized to promote positive mood states and high levels of eustress, thereby enhancing productivity [50]. Also, the proposed model allows for targeted interventions and personalized approaches to productivity enhancement. Individual workers can receive feedback and guidance based on their unique profiles, enabling them to understand and regulate their emotional states and behaviors for optimal productivity. By employing the extended productivity prediction model, organizations can take a proactive approach to enhancing productivity, resulting in higher job satisfaction, improved performance, and increased overall well-being among office workers [2].

## Figures and Tables

**Figure 1 sensors-23-08694-f001:**
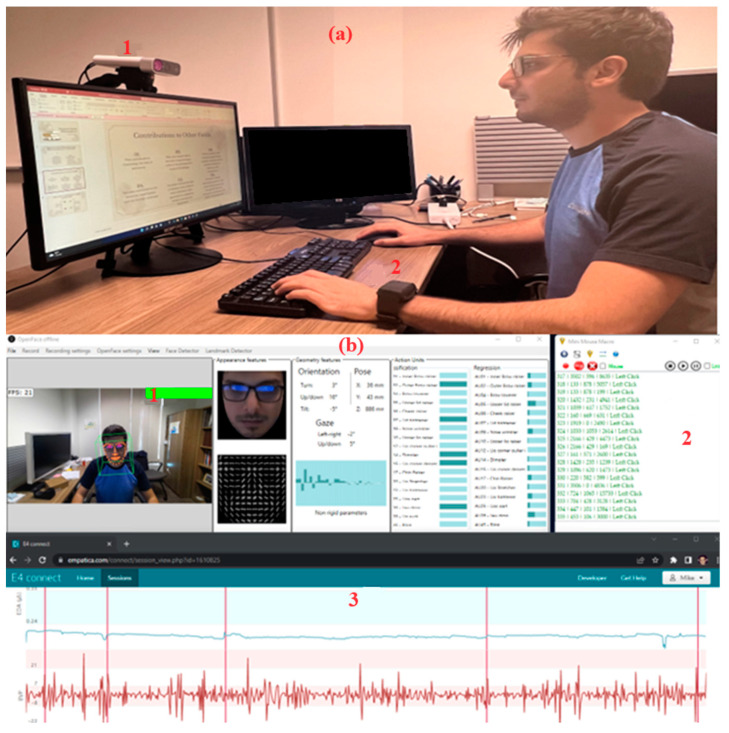
(**a**). Participant taking the experiment: (1) Kinect Camera and (2) Empatica E4 hand-wrist physiological sensor. (**b**). Data collection platform: (1) OpenFace application for facial feature extraction, (2) human–computer interaction application monitoring, and (3) physiological data monitoring.

**Figure 2 sensors-23-08694-f002:**
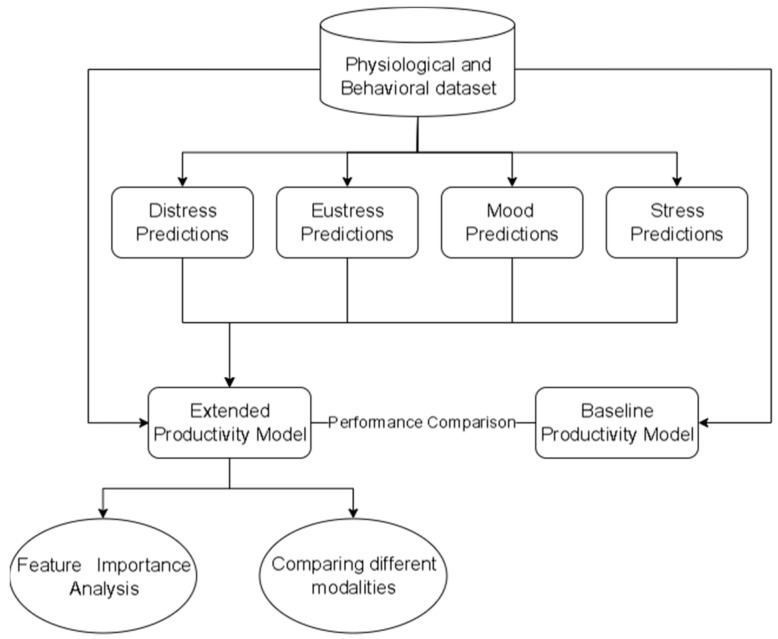
Overview of analysis.

**Figure 3 sensors-23-08694-f003:**
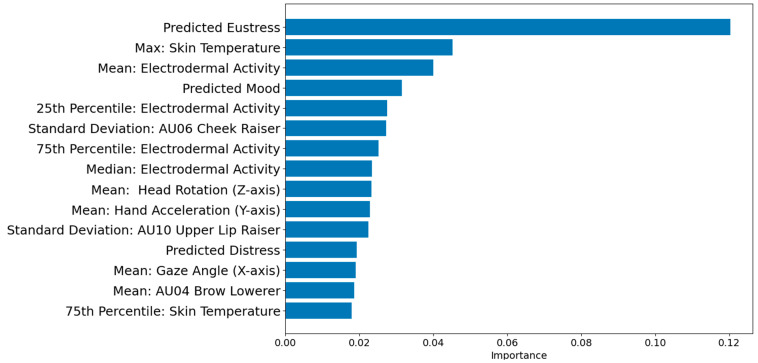
Feature importance for the extended productivity prediction model.

**Table 1 sensors-23-08694-t001:** Feature dataset.

Type (Number of Features)	Signal	Features Included
Physiological (34)	Electrodermal activity (EDA) Blood volume pulse (BVP) Skin temperature (ST)	Mean, standard deviation, median, minimum, maximum, 25th and 75th percentiles, slope fitted through the data
	Heart rate (HR) Heart rate variability (HRV)	Mean HR, standard deviation HR, minimum HR, maximum HR, rmsdd, LF peak, HF peak, LF power, HF power, LF/HF
Behavioral (48)	Facial action units (AUs) Head rotation Eye gaze direction	Mean, standard deviation
	Blink	Count
	Wrist acceleration	Mean, standard deviation
	Mouse right clicks Mouse left clicks Keyboard keystrokes	Count
Gender (1)	Female, Male	Binary

**Table 2 sensors-23-08694-t002:** Summary of mood, stress, eustress, and distress prediction models performance.

**Regression Analysis**				
**Algorithms**	**Mood**		**Stress**	
	**R^2^**	**MAE**	**R^2^**	**MAE**
Linear regression	0.06	10.36	0.09	14.74
Ridge regression	0.04	10.40	0.03	15.40
Lasso regression	0.01	11.13	0.01	15.68
Random forest	0.38	8.23	0.44	9.81
Gradient boosting	0.31	8.98	0.31	12.74
XGBoost	0.44	7.91	0.43	10.01
**Classification Analysis**				
**Algorithms**	**Eustress**		**Distress**	
	**Accuracy**	**F_1_-Score**	**Accuracy**	**F_1_-Score**
Logistic regression	0.79	0.65	0.53	0.68
Random forest	0.84	0.88	0.81	0.82
Gradient boosting	0.79	0.86	0.75	0.77
Decision tree	0.74	0.81	0.67	0.69
Support Vector	0.65	0.78	0.68	0.51
XGBoost	0.88	0.91	0.85	0.85

**Table 3 sensors-23-08694-t003:** Performance comparison between the baseline and extended productivity models.

Productivity Regression Analysis				
Algorithms	Baseline Model		Extended Model	
	R^2^	MAE	R^2^	MAE
Linear regression	0.25	21.59	0.27	16.42
Ridge regression	0.12	23.43	0.13	18.60
Lasso regression	0.10	24.16	0.15	22.60
Random forest	0.44	17.19	0.57	10.91
Gradient boosting	0.40	18.29	0.46	13.67
XGBoost	0.48	16.62	0.60	10.52

**Table 4 sensors-23-08694-t004:** Correlation analysis between predicted productivity and physiological and behavioral features.

	Pearson Correlation	*p*-Value
Physiological features		
Max: skin temperature	0.16	<0.001
75th percentile: skin temperature	0.16	<0.001
Mean: electrodermal activity	0.02	0.07
25th percentile: electrodermal activity	0.02	0.07
75th percentile: electrodermal activity	0.03	0.06
Median: electrodermal activity	0.04	0.04
Behavioral features		
Mean: wrist acceleration (*y*-axis)	−0.11	<0.001
Standard deviation: AU06 Cheek Raiser	0.08	<0.001
Mean: head rotation (*z*-axis)	0.06	<0.001
Mean: AU04 Brow Lowerer	−0.05	<0.001
Standard deviation: AU10 Upper-Lip Raiser	0.06	<0.001
Mean: gaze angle (*x*-axis)	0.13	<0.001

**Table 5 sensors-23-08694-t005:** Comparative analysis between wearable devices and workstation addons.

	E4 Empatica and H10 Polar		Kinect and Mini Mouse Macro	
	Performance Metrics	Best-Performing Algorithm	Performance Metrics	Best-Performing Algorithm
Mood	R^2^ = 0.43 MAE = 8.02	XGBoost	R^2^ = 0.23 MAE = 9.12	Random forest
Stress	R^2^ = 0.40 MAE = 11.17	Random forest	R^2^ = 0.27 MAE = 13.19	XGBoost
Eustress	Accuracy = 0.86 F_1_-score = 0.90	XGBoost	Accuracy = 0.81 F_1_-score = 0.86	XGBoost
Distress	Accuracy = 0.82 F_1_-score = 0.83	XGBoost	Accuracy = 0.77 F_1_-score = 0.76	XGBoost
Productivity: extended model	R^2^ = 0.56 MAE = 12.97	XGBoost	R^2^ = 0.50 MAE = 15.55	XGBoost

## Data Availability

The data presented in this study are available on request from the corresponding author. The data are not publicly available following the IRB guidelines associated with this study.

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
