# Peer review of "Predicting Office Workers’ Productivity: A Machine Learning Approach Integrating Physiological, Behavioral, and Psychological Indicators"

_sensors, 2023, doi:10.3390/s23218694_

Round 1
Reviewer 1 Report
This is a very interesting and well-structured study with potential extensions in different fields. The proposed methodology, in my opinion, is suitable and allows for deriving the proposed conclusions. However, I have some observations:
1. The studied population consists of young undergraduate and graduate students within a very narrow age range, with specific educational backgrounds, motivation, and high cognitive reserve. I believe it's important to emphasize this aspect as a limitation of the study or at least as a consideration for future research.
2. When comparing the performance of the workstation setups with wearable devices, it's also essential to highlight the differences in the variables being measured or, at the very least, how they are being measured. Since the experiment is the same in both cases, if the workstation devices were measuring the same variables as the wearable devices, perhaps the differences wouldn't exist. Please discuss this aspect in your manuscript.
3. Please clarify why, if in your study high levels of eustress improve productivity, you conclude that "Through intelligent lighting, temperature control, and ambient music, the office environment can be optimized to promote positive mood states and moderate levels of eustress, thereby enhancing productivity." Especially when you mention "moderate levels of eustress" to improve productivity.
Reviewer 2 Report
Overall, this study employs numerous features from various measures including physical, behavioral and psychological measures, making it interesting to the wide range of audience. This study is likely to have a high impact as the use of wearable sensors outperform the workstation addons, permitting the use of worker monitoring outside the office workstation.
Below are my specific points that I hope the authors would address to improve the manuscript.
1. The abstract gives a overview of the study. However, the modeling with machine learning is missing. It would be good if the authors could include a few phrases to tell which models are being used.
2. The introduction is very well-written. The context is logically connected and conveying. The research questions are clearly specified and very interesting. Nevertheless, the "related-work" section is too long. Particularly, the "traditional methods for productivity assessment" subsection is redundant to the first two paragraphs of the introduction. I believe this may be less of interest to the audience. I suggest the authors to shorten them or combine them with the context. This is an optional suggestion, please feel free to disagree.
3. In the "physiological and behavioral features for productivity assessment" subsection, I suggest that the authors rather focus on the sensors (the smartwatch and its parameters like HRV, EDA and ST, and camera), and software for human-computer interaction, instead of ML. Also, how about the BVP and accelerations, are there any studies mentioning them? This suggestion is also optional if the authors find it unnecessary.
4. In the "psychological features for productivity assessment" subsection, it is good that the authors give the definitions of stress, eustress and destress. It would be excellent if the authors could indicate how these manifestations are measured, as well as the mood.
5. In the methodology, your experiment lasted 70 minutes, dividing into a portion of 40-minute low-stress period and a 30-minute stressful one. Did you always ask participant to perform the tasks in the same order or was the order randomized? How long was the break?
6. In the data collection, there were several stressful elements in the protocol. Could you justify, element by element, why you chose them?
7. The stress and mood were subjective rating. Was any instruction given to the participants?
8. For those ratings, you used the increase and decrease from the baseline. Is there any proof that the scales were linear? Or was it your assumption?
9. For the features related to heart rate and heart rate variability, when you specified "mean" in Table 1, was in the mean of HR or the mean of RR? Similarly, the standard deviation, was it the SD of the HR, or SDNN? This also applies to the minimum and maximum.
10. Also, in the data processing, what does it mean by "slope fitted through the data" for the EDA, BVP and ST? I would assume a linear regression slope of the variables over time, but it would be better if you clearly add a short explanation.
11. The results are fascinating. Still, there are something you could improve. First, you could mention whether predicted mood, eustress and distress came from which model (XGBoost, RF, etc.) when you later feed them into the productivity model.
12. Even though the extended model significantly improved the prediction, you should also note that the correlation of 0.60 is rather moderate.
13. In the feature importance analysis, you used Pearson correlation, which has several assumptions. Have you checked with a scatterplot if the variables tend to have a linear relationship? Have you checked whether the data were normally distributed?
14. The feature importance analysis should be described in the method as part of your analysis plan.
15. About the effect of distress and eustress on productivity, could your interpretation be related to the nature of the study population?
16. One strength is definitely the high number of sample size. Meanwhile, there was a weakness when only university students are included. Would this group make the study generalizable to the working population?
My last comment is related to the inclusion, but if you think as inappropriate, simply ignore it. I would like mention that excluding people with vision problem and those using medication might have made the research inequitable, but I understand if you think it is not within your scope of work.
Reviewer 3 Report
This paper researches Machine Learning Approach Integrating Physiological, Behavioral, and Psychological Indicators. By introducing that the productivity of office workers has a direct impact on the success and growth of organizations and high levels of productivity result in increased efficiency, better time management and higher quality of work output which leads to increased profits and a positive financial return for companies, the authors proposed that the traditional method is outdated, modern wearable devices and software tools are better choices. The topic is emerging, unique, and interesting. The authors clearly described their method, and they have done plenty of experiments. It is obvious that this paper enlarged the research fields for machine learning approach and the contribution is significant, so I think this paper is acceptable. The author should make some revisions as follows:
1.The authors say “there exists a compelling rationale to investigate these data types' potential in predicting productivity among office workers” ,I suggest adding some detailed reasons here.
2. I suggest deleting the following sentence from the abstract:
While previous research suggests that VIF values may signify a noteworthy multicollinearity issue that requires attention, our results demonstrate that the extent of correlation in our model remains within acceptable limits, thus warranting no modifications.
3. I suggest considering revise the following sentences:
To our knowledge, no previous study has explored the use of physiological or behavioral features in assessing productivity among office workers, let alone incorporated psychological states. Consequently, these results serve as a foundational reference for future endeavors aiming to develop ML-driven solutions for predicting and monitoring productivity in an office environment.
4. Why do you mean the intricate interplay among these diverse domains contributes to the complex processes underlying productivity? You should introduce the expression more specifically.
5. The authors should check this paper carefully to reduce typo and grammar errors.
6. The reference for the correlation between higher wrist acceleration and productivity should be provided .
7. Despite relatively modest correlation coefficients (ranging from 0.02 to 0.04), the EDA-related features demonstrated a positive association with the predicted productivity. This finding could be attributed to…This sentences are hard to understand.
8. To ensure the generalizability and applicability of these results in real-world settings, a comprehensive longitudinal data collection approach becomes indispensable. The approach is not described clearly.
9. More related and advanced techniques should be reviewed or compared such as: Neural architecture search based on a multi-objective evolutionary algorithm with probability stack, Partial connection based on channel attention for differentiable neural architecture search, Improved differentiable architecture search with progressive partial channel connections based on channel attention, A self-adaptive mutation neural architecture search algorithm based on blocks
10. “Section 3 presents details about the process of collecting data, including the experimental setup, techniques for data cleaning and processing, and the training and evaluation of various ML algorithms. ” The steps of the experiment are too detailed.
11. The experimental results and analysis section is inadequate and, mostly, it consists of figures and tables without enough description and analysis. The whole analysis of the results can fit in one text column. Moreover, some detailed analysis could’ve been done to clarify some outcomes such as the inconsistency between training and test results.
This paper researches Machine Learning Approach Integrating Physiological, Behavioral, and Psychological Indicators. By introducing that the productivity of office workers has a direct impact on the success and growth of organizations and high levels of productivity result in increased efficiency, better time management and higher quality of work output which leads to increased profits and a positive financial return for companies, the authors proposed that the traditional method is outdated, modern wearable devices and software tools are better choices. The topic is emerging, unique, and interesting. The authors clearly described their method, and they have done plenty of experiments. It is obvious that this paper enlarged the research fields for machine learning approach and the contribution is significant, so I think this paper is acceptable. The author should make some revisions as follows:
1.The authors say “there exists a compelling rationale to investigate these data types' potential in predicting productivity among office workers” ,I suggest adding some detailed reasons here.
2. I suggest deleting the following sentence from the abstract:
While previous research suggests that VIF values may signify a noteworthy multicollinearity issue that requires attention, our results demonstrate that the extent of correlation in our model remains within acceptable limits, thus warranting no modifications.
3. I suggest considering revise the following sentences:
To our knowledge, no previous study has explored the use of physiological or behavioral features in assessing productivity among office workers, let alone incorporated psychological states. Consequently, these results serve as a foundational reference for future endeavors aiming to develop ML-driven solutions for predicting and monitoring productivity in an office environment.
4. Why do you mean the intricate interplay among these diverse domains contributes to the complex processes underlying productivity? You should introduce the expression more specifically.
5. The authors should check this paper carefully to reduce typo and grammar errors.
6. The reference for the correlation between higher wrist acceleration and productivity should be provided .
7. Despite relatively modest correlation coefficients (ranging from 0.02 to 0.04), the EDA-related features demonstrated a positive association with the predicted productivity. This finding could be attributed to…This sentences are hard to understand.
8. To ensure the generalizability and applicability of these results in real-world settings, a comprehensive longitudinal data collection approach becomes indispensable. The approach is not described clearly.
9. More related and advanced techniques should be reviewed or compared such as: Neural architecture search based on a multi-objective evolutionary algorithm with probability stack, Partial connection based on channel attention for differentiable neural architecture search, Improved differentiable architecture search with progressive partial channel connections based on channel attention, A self-adaptive mutation neural architecture search algorithm based on blocks
10. “Section 3 presents details about the process of collecting data, including the experimental setup, techniques for data cleaning and processing, and the training and evaluation of various ML algorithms. ” The steps of the experiment are too detailed.
11. The experimental results and analysis section is inadequate and, mostly, it consists of figures and tables without enough description and analysis. The whole analysis of the results can fit in one text column. Moreover, some detailed analysis could’ve been done to clarify some outcomes such as the inconsistency between training and test results.
Round 2
Reviewer 2 Report
The new version is better and more ready for publication.
Author Response
Reviewer comment: The new version is better and more ready for publication.
Response: We would like to express our gratitude to the reviewer for their thorough assessment of our manuscript and for their agreement to publish it.
Reviewer 3 Report
The layout of this paper is not well. The authors should adjust.
Author Response
Comment: The layout of this paper is not well. The authors should adjust.
Answer: The manuscript now follows the template provided by the Sensors Journal.